# HumanVid: Demystifying Training Data for Camera-controllable Human Image Animation

**Zhenzhi Wang**[1], **Yixuan Li**[1], **Yanhong Zeng**[2], **Youqing Fang**[2], **Yuwei Guo**[1],
**Wenran Liu**[2], **Jing Tan**[1], **Kai Chen**[2], **Tianfan Xue**[1,2], **Bo Dai**[3,2], **Dahua Lin**[1,2]
[1]The Chinese University of Hong Kong, [2]Shanghai Artificial Intelligence Laboratory,
[3]The University of Hong Kong
{wz122, ly122, gy023, tj023, tfxue, dhlin}@ie.cuhk.edu.hk
{zengyanhong, fangyouqing, liuwenran, chenkai}@pjlab.org.cn, bdai@hku.hk

## Abstract

Human image animation involves generating videos from a character photo, allowing user control and unlocking the potential for video and movie production. While recent approaches yield impressive results using high-quality training data, the inaccessibility of these datasets hampers fair and transparent benchmarking. Moreover, these approaches prioritize 2D human motion and overlook the significance of camera motions in videos, leading to limited control and unstable video generation. To demystify the training data, we present HumanVid, the first large-scale high-quality dataset tailored for human image animation, which combines crafted real-world and synthetic data. For the real-world data, we compile a vast collection of real-world videos from the internet. We developed and applied careful filtering rules to ensure video quality, resulting in a curated collection of 20K high-resolution (1080P) human-centric videos. Human and camera motion annotation is accomplished using a 2D pose estimator and a SLAM-based method. To expand our synthetic dataset, we collected 10K 3D avatar assets and leveraged existing assets of body shapes, skin textures and clothings. Notably, we introduce a rule-based camera trajectory generation method, enabling the synthetic pipeline to incorporate diverse and precise camera motion annotation, which can rarely be found in real-world data. To verify the effectiveness of HumanVid, we establish a baseline model named **CamAnimate**, short for Camera-controllable Human Animation, that considers both human and camera motions as conditions. Through extensive experimentation, we demonstrate that such simple baseline training on our HumanVid achieves state-of-the-art performance in controlling both human pose and camera motions, setting a new benchmark. Demo, data and code could be found in the project website: https://humanvid.github.io/.

## 1 Introduction

High-quality and highly controllable human image animation has significantly progressed as an emerging popular task [15, 28, 31, 74]. Imagine the possibilities of recreating iconic movie performances using just a single photo of the characters, capturing them from any desired angle. This technique has the potential to significantly impact video and movie production. In this study, we focus on animating characters from a single image, considering both human and camera motions as crucial factors for generating realistic human videos.

Despite recent advances [28, 88], human image animation presents two main challenges: the absence of a high-quality public dataset and the neglect of camera motions in human videos. Specifically, state-of-the-art approaches rely on private datasets for training similar models, underscoring the

38th Conference on Neural Information Processing Systems (NeurIPS 2024) Track on Datasets and Benchmarks.

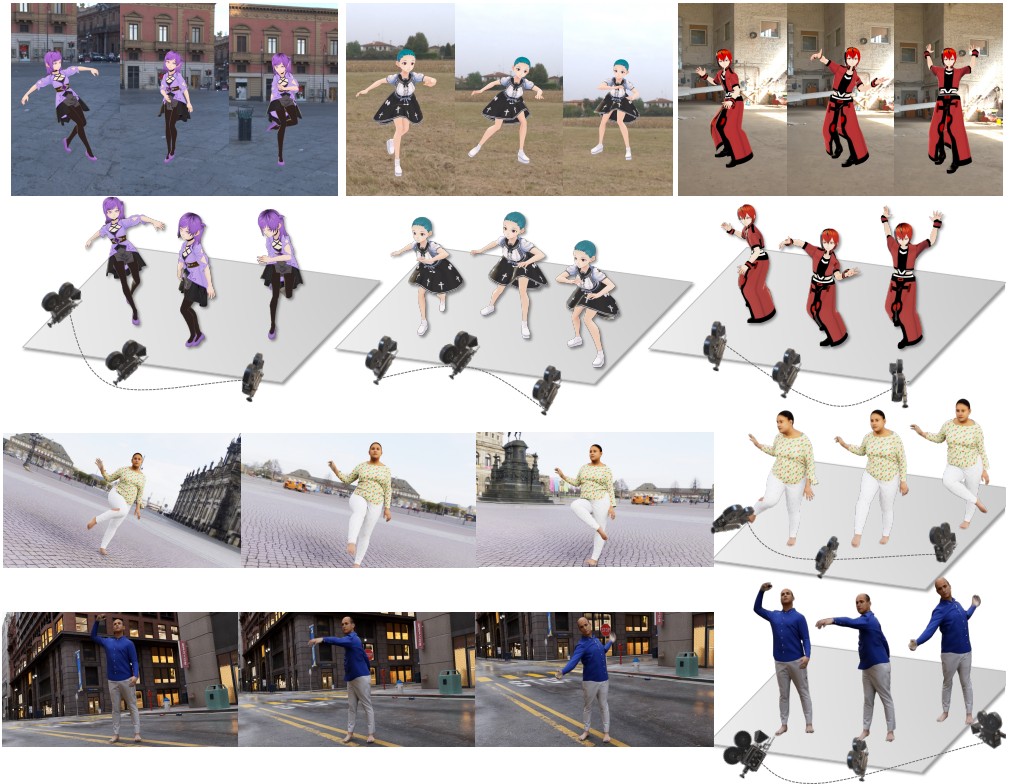

Figure 1: Illustration of controlling human poses and camera trajectories in our scalable synthetic data from anime characters (top) and human-like characters (bottom). Our synthetic videos has realistic human and background appearance and diverse camera trajectories.

importance of datasets in this field. However, these datasets remain inaccessible, while accessible alternatives like TikTok [29] and UBC-Fashion [80] possess limitations in scale and quality, hindering fair and transparency evaluation and community development. Furthermore, despite leveraging private datasets, these methods struggle to animate characters from new viewpoints with camera movement. They primarily rely on 2D pose extraction from static camera videos [75], neglecting the crucial aspect of camera motion. This design choice compromises video controllability and impedes high-quality character animation for complex motions.

To address the lack of a high-quality public dataset with accurate camera motion annotations, we introduce a synthetic dataset along with a high-quality real-world video dataset for human image animation. Our scalable pipeline allows us to create a large-scale dataset with precise annotations for both human and camera motions. Through extensive experiments, we validate the significance of this dataset combination in achieving high-quality and controllable human image animation.

To begin with, we compile a vast collection of real-world videos from diverse scenes on copyright-free internet platforms. This uncurated dataset often contains noise, such as frequent shot changes, user-generated special visual effects, occlusions and object motions. To ensure high-quality videos, we employ a carefully designed rule-based filtering strategy to ensure that pixel motions in our collected videos are exclusively resulted from human and camera motions. Additionally, we utilize SLAM-based methods [69, 62] for accurate camera trajectory extraction and a precise pose estimator [75] to extract human pose sequences. This results in a remarkable collection of over 20K human-centric videos in 1080P resolution. Experimental results demonstrate that training models exclusively on this curated video dataset achieve state-of-the-art performances.

Synthesizing diverse and high-fidelity human videos for animation is non-trivial. Existing 3D synthetic datasets for humans primarily focus on reconstruction or perception, resulting in limited appearance diversity, over-smoothed textures, and restricted camera motions [10, 76]. To overcome these limitations, we first augment our dataset with approximately 10K copyright-free 3D avatar assets. These assets undergo rigorous rigging using motions from motion-captured datasets [40] and open-source software [4], enabling a wide range of character shapes, appearances, and human

motions. Notably, to compensate for the limited camera motions observed in real-world videos, we introduce an innovative rule-based camera trajectory system, enriching the diversity of camera movements in training data. Specifically, multiple camera locations are randomly sampled for the keyframes throughout the space, and each camera is purposefully directed toward the human face. We connect and smooth these sampled camera keyframes to create natural camera movements similar to those found in professional videos and films. This design enables us to generate lifelike human videos with accurate annotations of human and camera motions. The illustration of our synthetic data in the rendering process is shown in Fig. 1. Our experiments conclusively demonstrate that utilizing our synthetic dataset significantly enhances animation, particularly regarding motion control.

To validate the collected dataset, we incorporate camera control [22] into a widely-used human video generation model that only considers pose condition [28]. Our main contributions are as follows,

- To the best of our knowledge, HumanVid is the first large-scale video dataset for human image animation. It contains both high-quality Internet videos with diverse appearance and synthetic videos with accurate human pose and camera pose annotations.
- We design a scalable rendering pipeline from Unreal Engine 5 that facilitates lifelike human video generation and provides accurate annotations of human and camera motions.
- Through extensive experiments, we validate the effectiveness of each dataset component, establish a new state-of-the-art, and create a comprehensive and transparent evaluation benchmark for the field.

## 2 Related Work

**Human Image Animation.** The task of human image animation aims to generate coherent human videos from a single image. To enhance controllability, the mainstream works in this field often employ explicit human skeleton representation, *e.g.*, OpenPose [13, 58, 72] and DensePose [19], as additional guidance. Early solutions are majorly developed upon GANs for image animation and pose transfers [14, 49, 55, 56, 57, 79, 84]. More recently, diffusion models (DMs) [24, 61, 42, 68] have been drawing attention from human image animation considering their remarkable success and high-quality results in image [51, 43, 48, 53, 7, 46] and video [11, 87, 59, 26, 25, 52, 77, 67, 21] synthesis. For instance, MagicDance [15] proposes a two-stage training strategy to disentangle the learning of appearance and human motion. Animate Anyone [28] utilizes a reference network to extract the appearance representation from the source image and adopts a motion module similar to AnimateDiff [21] to enhance temporal consistency. It also incorporates a lightweight pose guider to encode pose information to the pre-trained models. Similarly, MagicAnimate [74] adopts DensePose [19] as the motion representation and uses a ControlNet [81] to encode pose information. Champ [88] further introduces the SMPL [38] model sequence and the rendered depth and normal for better alignment. Though with remarkable visual quality, these works mostly adopt a static camera setting and do not consider camera viewpoint movement.

**Camera-aware Video Generation.** As a significant component in video and movie production, camera viewpoint movement determines the content dynamics and the overall feeling of the audience. While many works focus on guiding video generative models with structural signals [17, 77, 66, 83, 32, 20], less attention has been paid to controlling the pose/viewpoint of camera in generating videos [73, 78]. To control camera motion with reference videos, MotionDirector [85] proposes a dual-path LoRA [27] adapter to decouple the motion and appearance learning and can roughly control camera movements to produce a surrounding shot. For more precise control, MotionCtrl [71] directly injects the camera extrinsic matrix to the temporal attention layer in pre-trained text-to-video models and can precisely specify the camera viewpoint by providing camera poses at inference. CameraCtrl [22] further enhances the controllability by representing the camera pose with Plücker ray embeddings [60, 36]. CamViG [41] explores the camera control in token-based video generator [34] by introducing camera embedding as a new modality. Furthermore, JAWS [65] and Jiang *et. al.* [30] explore video generation via cinematic transfer from existing videos. To the best of our knowledge, our paper is the first to introduce camera control into the human-centric video generation task.

**Human Video Datasets.** Diverse and large-scale human-centric video datasets are essential for enabling human image animation tasks. For real-world datasets collected from the Internet, TikTok [29] provides 340 human-centric video clips from social media with diverse appearances and performances, while UBC-Fashion [80] contains 500 fashion video clips with blank backgrounds. To scale up the

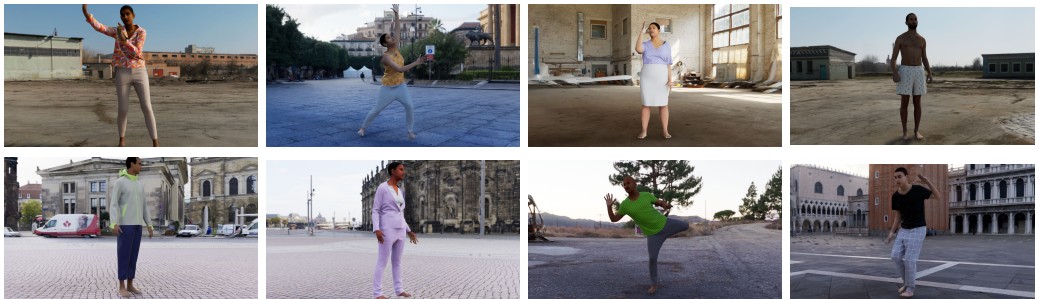

Figure 2: **Illustration** of the clothed SMPL-X characters rendered with diverse backgrounds.

dataset at a lower cost, several synthetic datasets have been proposed. SURREAL [64] generates over 6M realistic images of people rendered from 3D sequences of captured human poses, AGORA [44] renders from real human scans in diverse poses and natural clothing, and HSPACE [8] combines diverse individuals with different motions and scenes, obtaining the animation by fitting a human body model. GTA-Human [12] constructs a dataset with the GTA-V game engine, featuring a diverse set of subjects, actions, and scenarios. SynBody [76] includes 1.7M images with 3D human body annotations, covering diverse body models, actions, viewpoints, and scene styles. BEDLAM [10] renders people in realistic scenes and utilizes physics simulation to obtain realistically rendered clothing. While previous synthetic human datasets were designed for pose estimation or multi-view reconstruction, our work is the first to explore the use of synthetic data for video generation.

## 3 Dataset

Given that diffusion models typically require large amounts of data, we are pioneering the use of synthetic data in human video generation. While previous datasets [10, 76] only contain single-view image data or clips with basic camera movements (i.e. zoom-in, orbit), we show that accurate annotations, extensive scale and rich camera trajectories from synthetic data could be vital for generation. Our synthetic videos are rendered by Unreal Engine 5 (UE5) [3] or Blender [16]. To enhance the diversity of human appearance, we also curate human-centric internet videos from copyright-free platforms and leverage pose estimation methods [75] for automatic annotation. Both synthetic and internet data are fully scalable without any human supervision.

### 3.1 Synthetic Data Construction

The synthetic video data are rendered with one character moving in various 3D scenes using diverse camera trajectories. Consequently, constructing the synthetic data involves three key steps: character creation, motion retargeting, and 3D scene and camera placement.

#### 3.1.1 Character Creation

We create two types of characters in the diverse domains: (1) Human-like characters from SMPL-X [45] meshes and clothing. (2) Anime characters from user-uploaded assets. Diverse body shapes and skin textures, 3D clothing and textures are considered for highly varied human representation.

**Body shapes and skin tone.** For human-like characters, we sample body shapes from a diverse set of 271 body shapes with different BMI collected from the ARGOA [44] and CAESAR [50] datasets following Bedlam [10]. To reduce the gender and ethnicity bias, we use 50 female and 50 male commercial high-resolution skin albedo textures from Meshcapade [18] with seven ethnic groups.

**3D Clothing and textures.** To generate realistic human videos, it's crucial to have 3D clothing motions that are physically plausible and consistent with human body movements. For instance, the LSMPL-X representation from Synbody [76] adds a clothing layer to SMPL-X [45], but lacks realistic physics simulation for clothing motion, leading to unnatural movements in loose-fitting clothes like dresses. We collect 111 unique outfits, including T-shirts, sweaters, coats, jeans and skirts from Bedlam [10] dataset, and use commercial simulation software [1] to obtain realistic clothing deformations. On top of realistic meshes of human mesh and clothing from physics simulation, 1,691

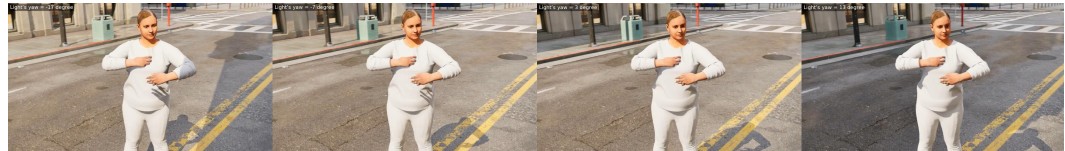

Figure 3: **Illustration** of the light condition control in the rendering process.

unique clothing textures are used for diverse clothing appearances. We shows examples of clothed SMPL-X characters in Fig. 2.

**Anime characters.** To enhance the diversity of characters in our synthetic data, we focused on VRoidHub [6], a platform designed for sharing 3D character models. Creators on VRoidHub have uploaded a vast array of intricate 3D character models, featuring diverse appearances, clothing styles, and hairstyles. From this rich repository, we manually selected 10K characters for rendering.

### 3.1.2 Motion Retargeting

Given the character assets, we transfer diverse motions to these characters by re-targeting motion data from various sources, including motion capture datasets [40] and open-source software Rokoko [4].

**SPML-X characters.** For human-like SMPL-X [45] characters, we sample human motions from large-scale motion capture datasets [40]. To enhance motion diversity, we sample based on motion annotations from [47], following the approach of Bedlam [10].

**Anime characters.** Conversely, anime character assets could have diverse skeleton lengths. We utilize the re-targeting software [4] to transfer existing motions to the anime character assets. The clothing and hair are treated as part of the body, so their motion is also determined by source motions.

### 3.1.3 3D Scenes and Camera Placements

**3D Scenes.** The realistic and diverse 3D scene backgrounds for synthetic video are constructed from about 100 panoramic HDRI images [2] or high-quality 3D scenes to cover both indoor and outdoor environment. We manually select panorama images with flat ground for characters to move on, while avoiding excessive scene components that might lead to unnatural human-scene interactions. We also exclude images with uniform visual patterns across different views, such as grasslands, deserts, or farms. The selected panorama backgrounds feature high-quality, complex texture details that highlight varying background textures from different camera angles.

**Camera Trajectory Design.** Unlike [10, 76, 44], our dataset highlights rich and diverse camera trajectories in human-centric videos. Each camera trajectory consists of a sequence of 6-DoF translations and rotations. We carefully design a rule-based camera motion generation pipeline to obtain diverse trajectories. This pipeline randomly sample camera locations adaptive to human positions and orientations in the keyframes, and use spline interpolation to get smooth camera locations and rotations in the whole sequence. Specifically, in each keyframe, we randomly sample camera locations within a semi-cylinder of radius $\in [3m, 5m]$ and height $\in [0.6m, 1.2m]$ in front of the human. Then, we set the camera orientation's yaw and pitch to point at the person. To create a more natural camera trajectory that smoothly follows the person, we adjust the camera's position by adding the human's position offset from the keyframe to the camera position in each frame. Finally, we also sample the roll of camera rotation $\in [-30°, 30°]$ in keyframes. Our design of camera keyframe sampling enables all types of camera trajectories, significantly enhancing the camera trajectory diversity and cinematic effect of human videos compared to existing video datasets.

**Rendering and Annotations.** We render the image sequences of SMPL-X characters using UE5 game engine and the built-in movie render function (Movie Render Queue) for high-quality images. The anime characters are rigged and rendered with blender. With our synthetic data source, a variety of ground-truth annotations, including camera trajectories, human skeletons, segmentation masks, depth maps and normal maps, could be obtained without manual efforts. Although our dataset is curated for human video generation, these ground-truth annotations could also be useful for other downstream applications. Thanks to the Unreal Engine, we could render photo-realistic videos with controllable human appearance and motion, scene, and light condition. As shown in Fig. 3, the light direction in the rendering process could be controlled from $-17°$ to $13°$.

Table 1: Comparison of our Internet and synthetic data size with existing datasets.

| Dataset | Clips | Frames | Resolution | Camera Pose | Human Pose |
|---|---|---|---|---|---|
| TikTok [29] | 340 | 93k | 604×1080 | Static | Fitting |
| UBC-Fashion [80] | 500 | 192k | 720 × 964 | Static | Fitting |
| IDEA-400 [37] | 12k | 2.5M | 720P | Static | Fitting |
| Bedlam [10] | 10k | 1.5M | 720P | Ground Truth | Ground Truth |
| Ours Real | 20k | 10M | 1080P | Fitting | Fitting |
| Ours Synthetic (SMPL-X) | 50k | 8M | 720P | Ground Truth | Ground Truth |
| Ours Synthetic (Anime) | 25k | 2M | 1080P | Ground Truth | Ground Truth |

Table 2: Statistics of the diversity of appearance, motion and scene in HumanVid.

| Dataset Split | #Subject | #Motion | #Scene | Avg. Clip Length |
|---|---|---|---|---|
| Internet videos | 24,012 | 24,012 | 19,688 (= #video) | 16.65s |
| Synthetic (SMPL-X) | 271 (body shapes) × 100 (skin textures) × 1,691 (clothings) | 2,311 | 100 (HDRIs) + 587 (3D scenes) | 6.34s |
| Synthetic (Anime) | 10K (anime assets) | 40 | 100 (HDRIs) + 93 (3D scenes) | 3.2s |

## 3.2 Internet Data Curation

To enhance the appearance diversity of synthetic videos, we collect real human-centric videos from copyright-free internet platforms [5], where the pixel motions in videos is only resulted from human skeleton motion or camera motion, without any object movements or background dynamics. We utilize the Pexels API [5] to scrape data based on around 100 keywords and employed a pose detector [75] to analyze the data. The pose detector focused on measuring the upper body keypoints' confidence, the ratio of the largest human bounding box over the frame $r$, the average number of humans present in each frame $n$, and the average motion (position offsets) of the keypoints $\Delta \bar{\mathbf{p}}$. With these statistics, we apply a specific filtering criteria: a) the human should occupy the main part of the image ($r > 0.07$); b) there should be few people ($n \leq 4$); c) there should be a noticeable motion in keypoints to remove static videos ($\Delta \bar{\mathbf{p}} > 0.01$); d) No exits, entrances or occlusions of individuals in videos ($| n - \text{round}(n) | < 0.01$). As a result, we collect around 20k high-quality, real human-centric video clips with various human and scene appearances.

**Camera Trajectory Estimation.** Reconstructing global camera trajectories from in-the-wild videos is a challenging problem. For the curated human-centric videos, we adopt TRAM [69] to utilize a SLAM method [62] for recovering camera extrinsic parameters from in-the-wild videos with explicit human movement. To ensure camera parameters are robust to dynamic humans, we employ a masking technique [33] that removes dynamic regions from both input images and dense bundle adjustment steps. To prevent major estimation errors, we configure the SLAM system to use only background features for camera motion estimation. To convert camera estimation to metric scale, we leverage semantic cues from the background by utilizing noisy depth predictions [9]. Consequently, we recover accurate, metric-scale camera motion that serves as an optimal camera condition for training diffusion models. When videos have backgrounds lacking texture and the SLAM system cannot accurately reconstruct camera motion, we treat these cases as having static cameras. Additionally, we filter out videos with very large rotations or translations, such as cycling, or those with sudden shot changes, as these videos fall outside the scope of human image animation.

**Statistics.** As shown in Tab. 1, our Real Internet dataset, with over 20k clips and 10M frames at 1080P resolution, significantly surpasses existing datasets like TikTok [29], UBC-Fashion [80] and IDEA-400 [37] in both size and resolution. Additionally, our synthetic dataset from SMPL-X and Anime characters, is 5× larger in scale compared to Bedlam [10], providing accurate camera and human pose groundtruth, and more diverse camera trajectories. For statistics of our HumanVid, Tab. 2 shows the statistics of human appearance, motion and scene.

## 3.3 CamAnimate

To validate our dataset's capability for animating humans with moving cameras, we propose a simple baseline for the camera-controllable human image animation task, named *CamAnimate*. By

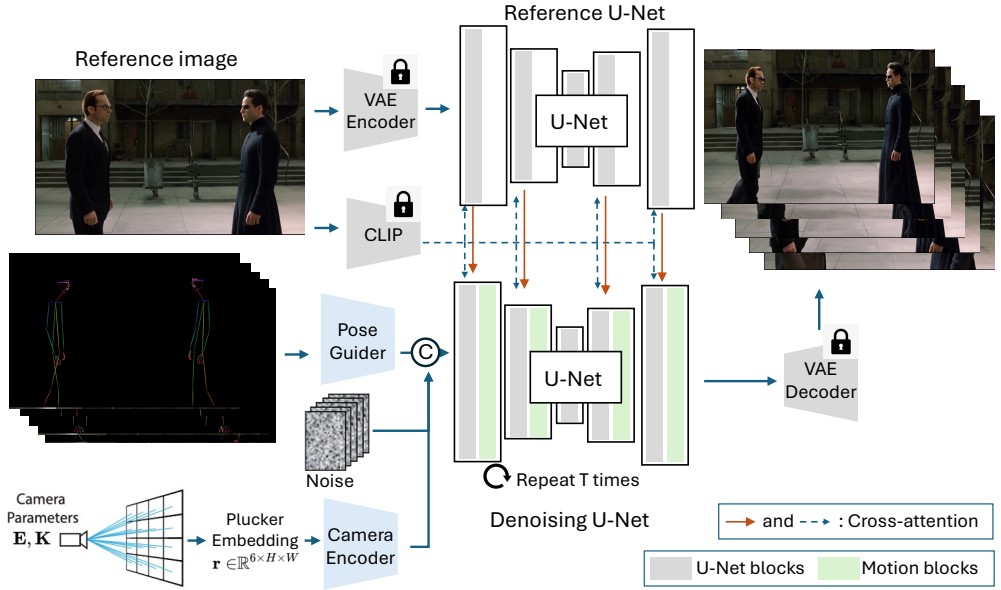

Figure 4: Illustration of the network architecture of our proposed CamAnimate.

leveraging CameraCtrl [22]'s advanced camera pose control and Animate Anyone [28]'s character animation framework, CamAnimate ensures consistent and high-quality human video generation with simultaneous human and camera movements. As shown in Fig. 4, it utilizes plücker embeddings to accurately parameterize camera trajectories and incorporates an additional camera pose encoder to encode camera information for the Denoising UNet via zero-convolution [81], while ReferenceNet and a pose guider maintain appearance consistency and pose controllability. By training our method on general human videos with both camera and human movement, these two types of motion can be decoupled in the network and learned by separate modules in an end-to-end manner. In addition to the moving camera setting, CamAnimate can seamlessly generate static-camera human videos. Please refer to Sec. A.1 for more implementation details of CamAnimate.

## 4 Experiments

**Evaluation Benchmark.** Due to the lack of a unified benchmark for previous methods, the testing protocols for each method have been varied significantly. However, when the form of the samples differs, the disparity between the reference image and the target image can vary greatly, leading to highly inconsistent results for the same method when inferring different reference and target images. Therefore, as the first large-scale dataset, we provide a unified testing protocol for human video generation. Specifically, we use the middle frame as the reference image, predict frames in the range [1,72] with a stride of 3, resulting in a sequence of 24 frames. Finally, we evaluate each video under this setting using PSNR [70], SSIM [70], LPIPS [82], FID [23], and FVD [63] metrics.

We use the last 40 videos from the TikTok dataset [29] out of a total of 340 videos as the test set for evaluation on static camera. For UBC-Fashion [80], we use the official split with 500 videos for training and 100 videos for testing. Additionally, we provide 40 videos each in both portrait and landscape orientations as a test set for evaluation on moving camera human video generation, sampled from our collected Internet videos. It is worth noting that SSIM, PSNR, and LPIPS are only reliable under static camera conditions without any human turning. For human videos with camera movements, these reconstruction metrics may not be trustworthy, as video generation inherently involves ambiguity, where we prefer generation metrics like FID and FVD.

### 4.1 Comparison with the State-of-the-Art

In this section, we compare our baseline model with previous state-of-the-art methods, namely Animate Anyone [28], Magic-animate [74] and Champ [88]. As animate anyone is not open-sourced,

Table 3: Comparison with SOTA on TikTok and UBC-Fashion dataset with static camera. † means its official implementation is not trained on UBC-Fashion's training set.

| TikTok Test Set | SSIM ↑ | PSNR ↑ | LPIPS ↓ | FVD ↓ | FID ↓ |
|---|---|---|---|---|---|
| Animate Anyone [28] | 0.752 | 16.971 | 0.288 | 935.6 | 52.26 |
| Magic-animate [74] | 0.748 | 17.890 | 0.270 | 876.0 | 56.84 |
| Champ [88] | **0.778** | 18.434 | 0.267 | 736.1 | 50.76 |
| Ours | **0.778** | **18.762** | **0.247** | **691.8** | **41.35** |

| UBC-Fashion Test | SSIM↑ | PSNR↑ | LPIPS↓ | FVD↓ | FID↓ |
|---|---|---|---|---|---|
| Magic-animate [74]† | 0.602 | 6.663 | 0.552 | 1583.9 | 118.76 |
| Animate Anyone [28] | 0.914 | 23.163 | 0.069 | 345.4 | 33.77 |
| Champ [88] | 0.922 | 25.269 | 0.057 | 269.4 | **27.35** |
| Ours | **0.929** | **25.921** | **0.049** | **256.6** | 29.30 |

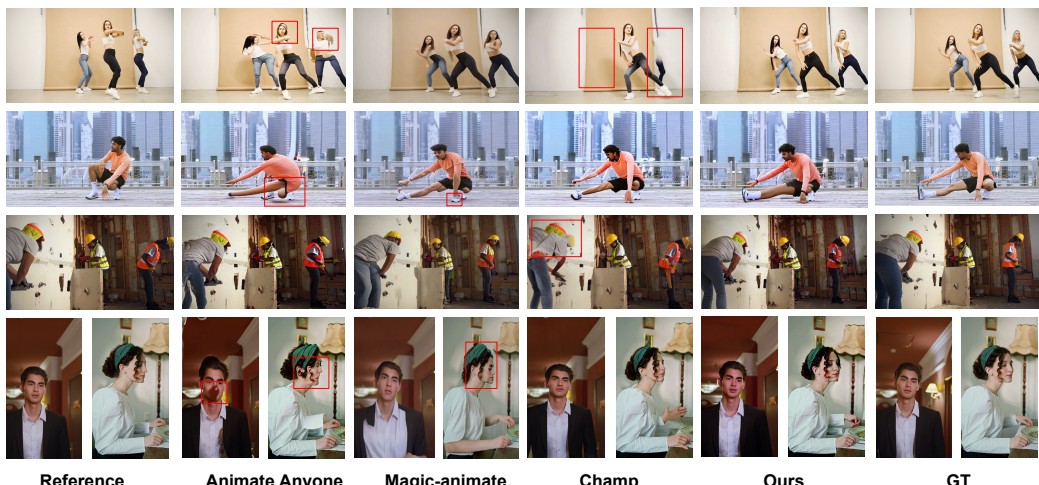

Reference    Animate Anyone    Magic-animate    Champ    Ours    GT

Figure 5: **Qualitative comparisons** with previous SOTA methods on the test set.

we use a third-party implementation[1]. We use the official implementations for other two methods. As shown in Tab. 3, although our method is trained on videos with moving cameras, it is still able generate high-quality static camera videos on TikTok and UBC-Fashion dataset and achieves best performance in almost all metrics due to our precise camera control ability. For human videos with camera movement on our test set shown in Tab. 4, previous methods commonly do not consider the camera condition, so they struggle to produce natural videos with camera movements. Such result could be observed from both reconstruction metrics like SSIM, PSNR, and LPIPS and generation metrics like FID and FVD. In our training set, we found that the performance could be further improved upon Animate Anyone's training recipe by increasing the effective batch size from $8 \times 8 = 64$ to $4 \times 8 \times 8 = 256$ through gradient accumulation, while maintaining 30,000 iterations in stage 1 training. This enhancement suggests our dataset's superior scale and diversity.

**User-study.** We also conduct a user-study to compare our method with previous methods, as shown in Tab. 5. We collect 2 videos from Tiktok test set and 8 videos from our test set, and compare our results with other three methods' results as a ranking question with 4 options. A total of 20 participants take part in our user study and we assign 3, 2, 1 points for the first, second and third method respectively. The final average score is normalized by total points, so the upper bound of this metric is 0.5. We conclude the average points and top-1 preference of each method in Tab. 5, which shows a dominate advantage (0.44 points and 0.73 top-1 preference) over previous methods due to their artifacts in appearance, human pose and camera movements.

**Qualitative comparisons.** In Fig. 5 , we show the qualitative comparison with previous SOTA methods, where the artifacts are highlighted by red boxes. We show that previous SOTA methods may have different artifacts when applied to our challenging evaluation test sets. For example, animate anyone suffers from inaccurate and low-quality appearances of humans. Champ suffers from missing

---

[1] https://github.com/MooreThreads/Moore-AnimateAnyone

Table 4: Comparison with SOTA on our collected human videos with camera movements.

| Landscape | SSIM ↑ | PSNR ↑ | LPIPS ↓ | FVD ↓ | FID ↓ |
|---|---|---|---|---|---|
| Animate Anyone [28] | 0.602 | 16.108 | 0.368 | 1248.4 | 97.74 |
| Magic-animate [74] | 0.543 | 15.567 | 0.361 | 1325.2 | 109.33 |
| Champ [88] | 0.653 | 15.028 | 0.426 | 1985.2 | 100.59 |
| Ours (1× batch size) | 0.641 | 18.008 | 0.309 | 960.1 | 77.73 |
| Ours (4× batch size) | **0.672** | **19.534** | **0.275** | **732.7** | **46.06** |
| **Portrait** | SSIM ↑ | PSNR ↑ | LPIPS ↓ | FVD ↓ | FID ↓ |
| Animate Anyone [28] | 0.613 | 15.514 | 0.379 | 1254.1 | 88.70 |
| Magic-animate [74] | 0.621 | 16.091 | 0.341 | 1418.8 | 123.94 |
| Champ [88] | 0.669 | 16.021 | 0.360 | 1316.9 | 84.59 |
| Ours (1× batch size) | 0.675 | 18.081 | 0.309 | 816.5 | 75.67 |
| Ours (4× batch size) | **0.678** | **18.939** | **0.303** | **792.2** | **54.02** |

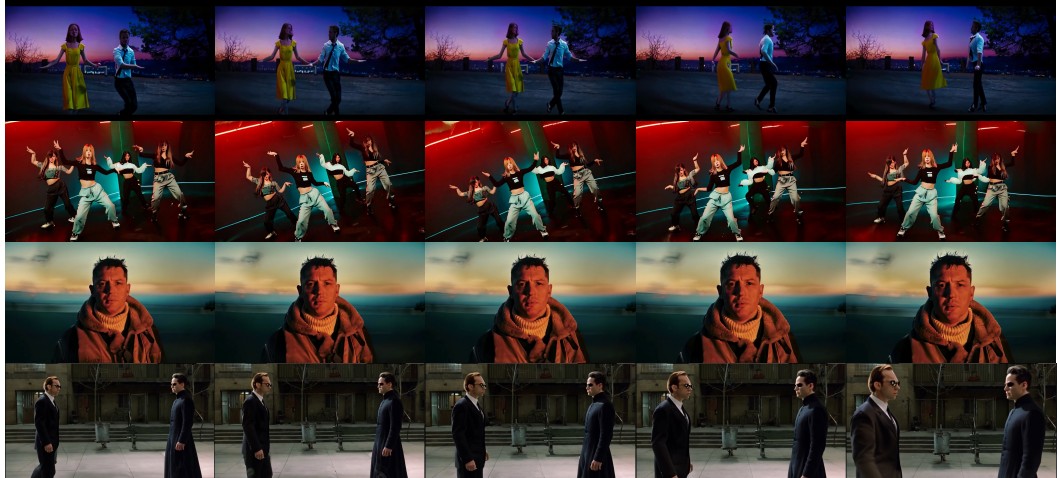

Figure 6: **Qualitative results** on in-the-wild videos with realistic camera movements.

3D skeletons due to the difficulty of estimating accurate 3D skeletons, especially in a crowded scene. Magic-animate is not able to correctly generate human face expressions due to the ambiguity of face representation of DensePose. Our method generates accurate facial expressions, body poses, and background motions that correspond to camera movements.

**Qualitative results on in-the-wild cases.** In Fig. 6, we further show our methods ability in generating in-the-wild videos with explicit camera movements. We show that we can achieve high-quality and visually pleasant videos that is close to the actual movies filmed by professionals. We hope that our dataset and the baseline method could serve as a solid base for generating movie-level human videos. Additional video results are available on the project page[2].

## 4.2 Ablation Study

**Contribution Decomposition of Dataset and Method.** We construct a variant that training the original Animate Anyone on HumanVid dataset without considering camera condition to verify that existing methods could not be directly applied to our dataset. Since most videos in HumanVid have moving cameras, it creates inherent ambiguity for models attempting to learn both camera movement and human movement simultaneously. Thus, the performance of CamAnimate and the variant trained on the same training set shows significant difference, as shown in Tab. 6. It indicates that the camera encoder in our CamAnimate is necessary for training human video generation models on HumanVid.

**Training Strategy.** Due to limited appearance diversity, synthetic data alone proves insufficient for training effective human image animation models, resulting in non-semantic, uniform textures that lack meaningful structure. While training with Internet videos enables appearance transfer and

---

[2]https://humanvid.github.io/

Table 5: User study on videos of Tiktok dataset and our test set.

| Method | Average Score | Top-1 Preference |
|---|---|---|
| Animate Anyone [28] | 0.171 | 0.10 |
| Magic-animate [74] | 0.133 | 0.03 |
| Champ [88] | 0.256 | 0.14 |
| Ours | **0.440** | **0.73** |

Table 6: Comparison with original Animate Anyone trained without camera condition.

| TikTok Test Set | SSIM ↑ | PSNR ↑ | LPIPS ↓ | FVD ↓ | FID ↓ |
|---|---|---|---|---|---|
| Animate Anyone [28] | 0.658 | 15.954 | 0.337 | 1133.1 | 53.65 |
| Ours | **0.778** | **18.762** | **0.247** | **691.8** | **41.35** |

Table 7: Comparison of training strategies on different data parts.

| TikTok Test Set | Stage 1 w/ Syn. Data | Stage 2 w/ Syn. Data | SSIM↑ | PSNR↑ | LPIPS↓ | FVD↓ | FID↓ |
|---|---|---|---|---|---|---|---|
| Variant 1 | × | × | 0.677 | 15.957 | 0.333 | 1066.9 | 53.08 |
| Variant 2 | ✓ | ✓ | 0.734 | 17.339 | 0.287 | 980.3 | 56.32 |
| Ours | × | ✓ | **0.778** | **18.762** | **0.247** | **691.8** | **41.35** |

pose control capabilities, camera control remains challenging. This limitation primarily stems from the difficulty in extracting accurate camera trajectories from Internet videos using COLMAP [54] or SLAM methods [62], as moving subjects often obscure static backgrounds. To address this, we leverage our synthetic data to enhance camera control accuracy and overall visual quality through a two-stage training strategy. In stage 1, we finetune the ReferenceNet, Pose Guider, DenoisingNet, and Camera Encoder. In stage 2, we focus solely on finetuning the Camera Encoder and Temporal Modules in DenoisingNet. This approach effectively combines high-quality Internet videos for human appearance modeling with synthetic videos for improved camera control. As shown in Tab. 7, our final model outperforms both Variant 1 (using only Internet videos) and Variant 2 (using both video types for both training stages).

**Camera Trajectory Evaluation.** Due to the inaccurate camera trajectories estimated from our gener-ated videos, it is difficult to obtain accurate camera-related metrics (Translation Error and Rotation Error) used in CameraCtrl [22]. The primary reason is that the generated videos in CameraCtrl only have static scenes and moving cameras, where COLMAP [54] is proven to be effective in extracting accurate camera trajectories. However, because we have moving people and moving cameras, such Structure-from-Motion methods will fail in our setting. Thus, we leave it to our future work to quantitatively verify the camera control ability.

## 5 Conclusion

In this paper, our study addresses the significant challenges in the field of human image animation by introducing a novel combination of a high-quality real-world video dataset and a meticulously crafted synthetic dataset. Our proposed dataset not only enhances the visual quality and controllability of human animations, but also introduces a new benchmark for camera control in human videos. Without bells and whistles, our proposed simple baseline demonstrate superior performance when it is trained on our combined dataset, particularly in scenarios involving complex human and camera motions. We believe our dataset establishes a foundation for transparent and comprehensive evaluations in this field, facilitating future advances in video and movie production.

**Limitations.** The annotation of our Internet data heavily rely on pose estimation [75, 19] and SLAM [69, 62] methods, which could introduce noises into the camera and pose annotations. The number of asset and background scenes in the synthetic part is limited when we compare it to real videos. The rendering quality is also worse than the real videos captured by professional cameras.

**Broader Impacts.** Our dataset and baseline method are highly effective at creating realistic human videos. Nonetheless, it's important to recognize that improvements in generative model technologies could lead to the creation of realistic deepfakes, which may be misused to spread misinformation.

**Acknowledgment.** This project is funded in part by Shanghai Artificial Intelligence Laboratory, CUHK Interdisciplinary AI Research Institute, and the Centre for Perceptual and Interactive Intelligence (CPII) Ltd under the Innovation and Technology Commission (ITC)'s InnoHK. This project is supported by RGC Early Career Scheme (ECS) No. 24209224 and CUHK Direct Grants (RCFUS) No. 4055189. We would like to thank Xuekun Jiang for helpful discussion.

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

# A Appendix

## A.1 Implementation Details of CamAnimate

We use the checkpoint of Stable Diffusion 1.5 [51] to initialize the Denoising UNet and ReferenceNet, and use the weights of ControlNet [81] on OpenPose [13] to initialize the Pose Guider. The camera encoder weights from CameraCtrl [22] are used to initialize our camera encoder. We mix train horizontal and vertical videos with a resolution of (long side, short side) $= (896, 512)$, i.e., for horizontal videos $(w, h) = (896, 512)$ and for vertical videos $(w, h) = (512, 896)$. Each batch only samples either all horizontal or all vertical videos, and the choice between horizontal and vertical videos is made randomly between batches. The setting of such resolution is according to the GPU memory in our experiments, i.e., maximum GPU memory usage with such pairs of batch size and resolution in both stages. We empirically find that such resolution could achieve a balance of visual quality and computational cost. In the first stage, we train all network parameters using a batch size of 8. In the second stage, we freeze the denoising UNet, reference UNet, and pose guider, and only train the camera encoder and motion module. The motion module in the second stage is initialized with the weights from AnimateDiff [21] V3. The frame rate is set to 24, and the batch size is 1. We use 8 NVIDIA A100 GPUs for all training stages and 1 NVIDIA A100 GPU for testing. The first and second stages are trained for 30,000 and 10,000 iterations, respectively, with a learning rate of 1e-5 and AdamW optimizer [39]. Our camera embedding representation is the Plücker embedding from CameraCtrl, which is computed from the camera's intrinsic and extrinsic parameters. For real internet data, the intrinsic parameters are set using a heuristic value, while the extrinsic parameters are obtained using SLAM [62] methods. For synthetic data, the intrinsic and extrinsic parameters are directly exported. For the $4 \times$ batch size setting in stage 1, we use 8 NVIDIA A100 GPUs with gradient accumulation step as 4, or 32 NVIDIA A100 GPUs with gradient accumulation step as 1. The batch size per GPU is always 8 due to the GPU memory limits. We also train the network for 30,000 iterations in $4 \times$ batch size setting and use the learning rate as 1e-5.

## A.2 More Details about Rendering of Synthetic Data

**Flow chart of the synthetic data creation pipeline.** In Fig. 7, we show a detailed flow chart to help readers better understand our rendering process.

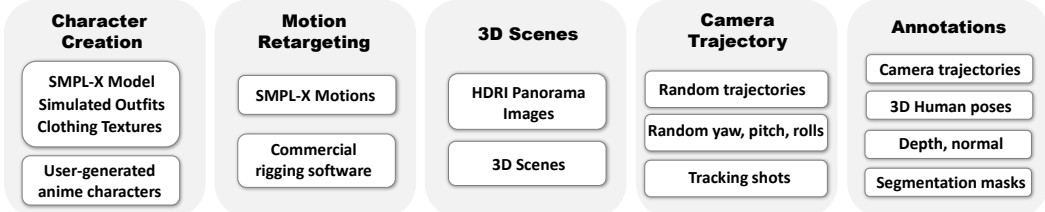

Figure 7: **Flow chart** of our synthetic data creation pipeline.

**Process of anime characters creation.** In Fig. 8, we show the illustration of video rendering process in creating anime characters.

## A.3 Camera Movement Comparison

As our goal is to train a controllable video diffusion model conditioned on camera trajectory and human pose, we think that let the model be trained on a large range of camera movements (i.e., offsets) in all 6 dimensions (XYZ, roll, yaw, pitch) is important. It is also the major motivation to construct the synthetic part of data in our HumanVid dataset. In Fig. 9, we show statistics of camera pose offsets of these 6 dimensions. Quantitative results shows that the frame-wise camera pose offset (i.e., the camera movement of next frame) of synthetic data part shows larger distribution than the real data part. It illustrates that our camera trajectory design produces large enough camera movement for modeling natural camera movements in the Internet videos. Besides, experimental results on in-the-wild cases in inference shows that model trained on such camera trajectories could follow camera trajectories in real world videos very well.

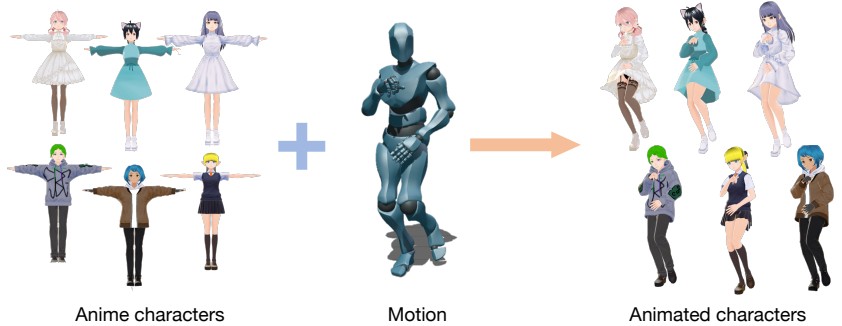

Anime characters          Motion          Animated characters

Figure 8: **Illustration** of the process of creating our synthetic assets in anime characters.

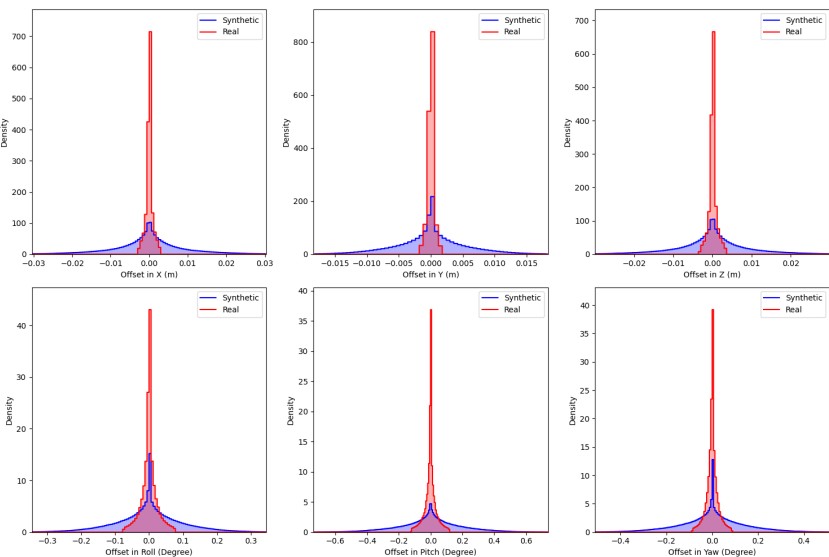

Figure 9: **Statistics** of camera movement strength comparison between synthetic data and real data in the space of location and rotation.

Besides, the current trajectory of camera is achieved by setting two keyframes: the first frame and the last frame. Although two keyframes could already enable the camera movement to cover a large region, it is also possible to create synthetic videos using three keyframes and five keyframes in our rendering pipeline. Such cases could be found in the supplementary materials or the project website.

## A.4    Visualization of Human Mask, Depth and Normal

In Fig. 10, we show the example of paired data of RGB image, human segmentation mask, depth map and normal map. It is achieved by putting SMPL-X characters [10] at a random place of a large-scale city scene dataset [35]. The scene and human are jointly rendered by Unreal Engine. As the major focus of our dataset is not 3D information, we only shows that our rendering pipeline is able to produce human mask, depth and normal, yet our dataset will not provide them.

## A.5    Visualization of Human Keypoints Tracking

While our rendering pipeline primarily focuses on generating synthetic videos with human pose and camera parameters, it can also perform point tracking functionality similar to PointOdyssey [86]. As demonstrated in Fig. 11, we present a preliminary example of human keypoint tracking, where we specifically track human keypoints rather than all scene or object points. Our rendering pipeline has the potential to support point tracking data generation with some additional development.

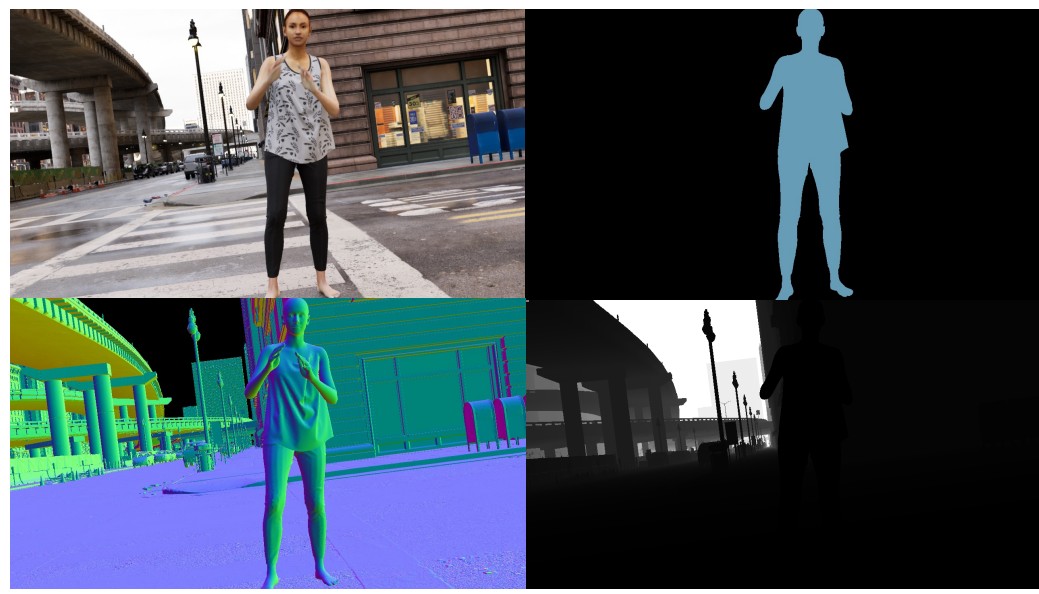

Figure 10: **Illustration** of human segmentation mask (top right), normal map (bottom left), and depth map (bottom right) in our synthetic data.

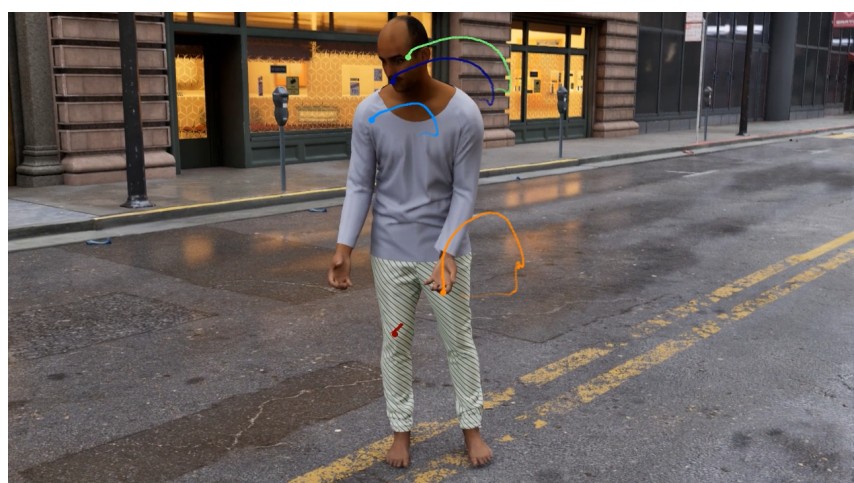

Figure 11: **Illustration** of human keypoints tracking by leveraging our synthetic data creation pipeline.

## A.6   Details of User Study Questionnaires.

We show the description of our user study as follows.

```
We tested 10 samples, each using 4 different character animation methods.
Please make a comprehensive judgment based on the following factors with
descending importance.
1. Appearance of the character (compared to the first frame) and naturalness
of the movement;
2. Smoothness of the background when the camera is moving;
3. Stability of the image and minimal color flickering.
Please rank the 4 animations for each sample in order of preference.
```

Then, for each question, we provide the image of first frame, and 4 videos generated by 4 methods. The order of videos are randomized for each question to ensure the fairness of user study.

# B Dataset Documentation

This dataset is designed to train camera-controllable human image animation models. Once trained, these models can also perform previous setting of static camera human image animation by passing camera parameters that represent a static camera to the model. For the latest documentation, please refer to the code in Github[3].

To train a human image animation model on HumanVid, follow these steps:

- Download internet videos using the video URLs provided in the text files. Additionally, download the synthetic videos and camera parameters from the cloud drive.
- Extract human poses from both internet and synthetic videos using DWPose [75].
- Extract camera trajectories for internet videos using Droid-SLAM [62] following TRAM [69], i.e., using Segment Anything [33] to mask out humans before SLAM.
- Compile all valid camera parameters, RGB videos, and human pose videos into a JSON file as meta-information.
- Download pretrained checkpoints for CLIP, SD 1.5, Animatediff, CameraCtrl, etc.
- Train the first stage of CamAnimate, which involves learning image-level appearance and pose control. The camera encoder at the image level is also trained in this stage.
- Train the second stage of CamAnimate, which focuses on learning video-level motion representations and camera control capabilities.

**Camera Parameter Format.** Following Droid-SLAM [62], we use the TUM camera format[4] to save the camera parameters. Each data line adheres to the following structure: 'timestamp tx ty tz qx qy qz qw'. The timestamp is a floating-point number representing the elapsed seconds since the Unix epoch. The values tx, ty, and tz (float numbers) denote the position of the optical center of the color camera relative to the world origin as defined by the motion capture system. The values qx, qy, qz, and qw (float numbers) represent the orientation of the optical center of the color camera in the form of a unit quaternion, again relative to the world origin as defined by the motion capture system. For camera parameters of internet videos estimated by Droid-SLAM, these are camera-to-world parameters. For synthetic data exported by rendering engines, we set these as world-to-camera parameters. The last two columns of camera parameters for synthetic data represent camera intrinsics, which is the normalized focal length according to the RealEstate10K dataset [5], i.e., (focal_length / sensor_width, focal_length / sensor_height).

# C Author Statements

We use the CC-BY 4.0 [6] license for our HumanVid dataset. We promise that we bear all responsibility in case of violation of rights.

## C.1 Licenses

In our dataset, we use assets from the following sources, and we list their licenses below.

**BEDLAM** [10]: The licensor of Bedlam grants us to use the asset of SMPL and SMPL-X parameters, 3D body and clothing meshes, 2D textures, and scripts for academic usage, according to the statement of their website [7]. We use such assets to generate SMPL-X character videos in our HumanVid dataset.

**Pexels** [5]: Pexels is a copyright-free website that grants us to use their videos for free according to the statement of their website[8]. We use videos from Pexels as the internet videos part in our dataset.

---

[3] https://github.com/zhenzhiwang/HumanVid
[4] https://cvg.cit.tum.de/data/datasets/rgbd-dataset/file_formats
[5] https://google.github.io/realestate10k/download.html
[6] https://creativecommons.org/licenses/by/4.0/
[7] https://bedlam.is.tuebingen.mpg.de/license.html
[8] https://www.pexels.com/license/

**VroidHub** [6]: VroidHub is a platform that collects user-generated contents. The description of license for each 3D asset is in this website[9]. All the 3D avatars we used in our dataset clearly show the permission of usage in their individual websites.

**Rokoko** [4]: Rokoko is a open-source blender addon to do motion retargeting on animatable 3D avatars. Its license is in this Github Repo [10], which grants us to use it in our dataset.

# D    Maintenance Plan

We will maintain the synthetic videos of our dataset by a cloud drive. For internet videos, we will provide video links to download raw videos from the website, as we could not redistribute these videos.

---

[9]`https://hub.vroid.com/en/license?allowed_to_use_user=everyone&characterization_allowed_user=everyone&corporate_commercial_use=allow&credit=necessary&modification=disallow&personal_commercial_use=nonprofit&redistribution=disallow&sexual_expression=allow&version=1&violent_expression=allow`

[10]`https://github.com/Rokoko/rokoko-studio-live-blender/blob/master/LICENSE.md`

