# OpenReview forum: "HumanVid: Demystifying Training Data for Camera-controllable Human Image Animation"
_NeurIPS.cc/2024/Datasets_and_Benchmarks_Track — NeurIPS 2024 Track Datasets and Benchmarks Poster_

### Official Review · Reviewer_tuQT · 2024-07-21

**Rating:** 7
**Confidence:** 5
**Clarity:** yes

**Review:**

The writing quality, clarity, originality, and significance of the paper are all good, making it easy to understand and follow.

**Strengths:**

1. The motivation of the paper is reasonable. Real-world data cannot accurately obtain high-quality camera motions, and the current real-world data is difficult to acquire due to strict copyright issues.

2.This dataset offers many unique features, such as high resolution and a large volume of data, as shown in Table 2.

**Additional Feedback:**

none

**Correctness:**

The data construction and evaluation sound good; however, the data statistics and comparisons need further improvement.

**Documentation:**

The work only provides some samples; it would be better if a download link for all the data could be provided. I am interested in the overall distribution of the dataset. The maintenance plan appears to be overly simplistic; a dedicated dataset page introducing the unique aspects and data distribution of this work seems necessary. This could enhance the impact of the work.

**Limitations:**

see Opportunities For Improvement

**Opportunities For Improvement:**

1. In fact, synthetic data has many advantages over real data. This paper appears to explore only a portion of these capabilities.
For example, while synthesizing masks, depth, and normal maps, it is also possible to consider outputting 2D and 3D pixel motion trajectories, as done in PointOdyssey. I understand this is not the main focus of the work, but if the corresponding UE5 project files could be open-sourced, it would be helpful for other tasks as well.


[1] PointOdyssey: A Large-Scale Synthetic Dataset for Long-Term Point Tracking

2. Additionally, some detailed statistics about the synthetic data seem to be missing, such as the average duration of video clips, the number of motion styles, and the average number of objects (humans) appearing in each video. As a dataset-focused work, more comprehensive data statistics and comparisons are needed to provide the reader with a complete understanding of the dataset.

**Relation To Prior Work:**

this work discussed the difference from previous works.

**Summary And Contributions:**

The paper proposes a high-quality dataset tailored for human image animation, which combines crafted real-world and synthetic data. It gathers 2,300 copyright-free 3D avatar assets to augment existing available 3D assets. Additionally, the work introduces a rule-based camera trajectory generation method, enabling the synthetic pipeline to incorporate diverse and precise camera motion annotation.

---

> ### Author Rebuttal · Authors · 2024-08-17
>
> **Q1: Output 2D and 3D pixel motion trajectories, as done in PointOdyssey**.
>
> **A1:** Thank you for your insightful suggestion. While this feature is not the primary focus of our current work, I believe it will be highly beneficial for other tasks. We have carefully examined PointOdyssey's approach: they first render RGB, depth, normal, and mask data. Then, they automatically export the mesh of both the static scene and dynamic characters in OBJ format. Finally, they cast random points on the mesh, calculate visibility using depth information, and project these visible points onto the 2D image based on the camera pose to obtain 2D point tracking.
> At present, our pipeline has accomplished the first step. The third step could be readily incorporated into our work. For the second step, additional engineering implementation is required to automatically export the scene from Unreal Engine 5. We provide a preliminary demo of point tracking for human in the 'For_Rebuttal/Point_tracking' subfolder in our [cloud drive](https://mycuhk-my.sharepoint.com/:f:/g/personal/1155189552_link_cuhk_edu_hk/EjLaZjITC79Cvqq3uyimlnoB9vSvAhz35b8Fhykt-QHiNw) (password: humanvid_cuhk). We will make all our UE5 project file publicly available and let it be user-friendly if others want to develop the point tracking function upon our UE5 project.
>
> **Q2: Some detailed statistics about the synthetic data seem to be missing, such as the average duration of video clips, the number of motion styles, and the average number of objects (humans) appearing in each video.**
>
> **A2:** Thanks for your suggestion. We show the detailed statistics in the following Table.
>
> | Dataset Split | #Subject | #Subject per video | #Motion | #Scene | Avg Clip Length |
> | --- | --- | --- | --- | --- | --- |
> | Internet videos | 24012 | 1.21 | 24012 | 19688 (= #video) | 16.65s |
> | Synthetic (SMPL-X) | 271 (body shape) X 100 (skin textures) X 1,691 (clothing textures) | 1.0 | 2311 | 200 (HDRI Images) + 587 (locations from a city-scale outdoor scene) | 6.34s |
> | Synthetic (Anime) | 2,381 anime assets | 1.0 | 40 | 200 (HDRI Images) +  93 (3D scenes by designers) | 3.2s |
>
> **Q3: The work only provides some samples; it would be better if a download link for all the data could be provided.**
>
> **A3:** Thanks for your suggestion. We have provided >10,000 synthetic data and >18,000 urls of real data in the cloud drive link provided in Line 77 of our Appendix in Supp. Mat.. We will provide all the data along with this paper’s final version.
>
> **Q4: I am interested in the overall distribution of the dataset. The maintenance plan appears to be overly simplistic; a dedicated dataset page introducing the unique aspects and data distribution of this work seems necessary.**
>
> **A4:** The data statistics has been shown in Q2. Since the evaluation of video generation model does not require human annotations, we do not need to maintain a test server for evaluating the performance on the hidden test set. Our major maintenance plan is to ensure that most of people could download our videos, camera files and human pose files. Besides, we will make our rendering scripts publicly available and maintain such code for others to develop their synthetic data based on ours. We will add a dataset page introducing the unique aspects and data distribution in the final version.

---

> > ### Comment · Reviewer_tuQT · 2024-08-18
> >
> > Thank you for the author's response. This dataset appears to be a very promising contribution to the community.
> >
> > After discussing with the reviewers, I will consider further increasing my score. Thank you.

---

> > > ### Author Response · Authors · 2024-08-18
> > >
> > > Thank you for your thoughtful review. We sincerely appreciate your valuable suggestions, which have greatly improved the quality of this work.

---

### Official Review · Reviewer_Egiq · 2024-07-22
**A dataset and partially new task for 2D human video generation with camera movement**

**Rating:** 7
**Confidence:** 3
**Correctness:** The dataset construction and evaluati…
**Clarity:** The paper is well written.

**Review:**

# Pros.
1. The paper is well organized. It also provides comprehensive reference related to dataset source and implementation details.
2. The task of 2D human animation generation with camera motion is interesting. The dataset proposed could be useful for the current trending 2D human animation works.
3. The construction of the synthetic dataset make sense. The quality of synthetic datset from SMPL-X is overall good.

# Cons.
1. The author mentioned that model trained only on synthetic dataset cannot generate meaningful results due to limitation of dataset diversity. While the length of the synthic dataset is almost equal to the real dataset. I wonder if the synthetic dataset is constructed with duplicate information. Can you show some metrics related to the diversity or the distribution for the real-world dataset and synthetic datet?
2. The rendering quality for synthetic (Anime) data is not good. The background looks like stickers but not 3D and the character texture is rough. I suspect that this low-quality data is not helpful for good performance.
3. It seems to be not very clear about the dataset split used to train the model for comparison. I'm not very convinced that the animation performance with static camera will improve by training with additional low-quality synthetic data.

Overall, the author devote efforts in building a large-scale 2D human animation dataset, especially the synthetic dataset. The quality of part of the synthetic (SMPL-X) dataset is good while the other part (Anime) is not. I hope the author could conduct more analysis of the dataset especially the synthetic one to demonstrate its contribution.

**Strengths:**

See the Pros.

**Additional Feedback:**

NA

**Documentation:**

The dataset is well documented.

**Ethics:**

The real-world dataset from pexels involves high-quality human-centric video.

**Limitations:**

The author has discussed that the limitation is mainly from the quality of the camera pose and human motion estimation. From my view, the error of the camera pose may affect more on the performance. Can you try to quantitatively show the overall error of the camera pose?

The quality of the synthetic dataset is still limited. It should also be discussed in the paper.

**Opportunities For Improvement:**

1. I wonder if the light and shading effects are considered for the synthetic (SMPL-X) dataset. The quality of the synthetic dataset needs to be improved.
2. The trajectories of the camera seem not to be diverse enough. Is it possible to generate the camera trajectory more naturally?

**Relation To Prior Work:**

The relation to prior work is discussed.

**Summary And Contributions:**

# Summary
This paper presents HumanVid, a large-scale 2D human animation dataset that combines real-world and synthetic data with camera annotations. For the real-world data, it adopts rule-based filtering to acquire high-quality video and off-the-shelf methods to estimate the human motion and camera pose. For synthetic data, it utilizes 3D character models and 3D scenes from various resources. It generates camera motion randomly to render synthetic video. To validate the dataset, a partially new task and benchmark for 2D human animation generation with a moving camera is presented. The model trained on the proposed dataset shows superiour performance.
# Contributions
1. To the best of my knowledge, it's the first open-source large-scale for 2D human animation generation with camera movement.
2. It potentially provide a benchmark for the trending task of 2D human animation and a partially new task with camera motion.
3. It build a large-scale synthetic dataset with camera, depth, motion annotations.

---

> ### Author Rebuttal · Authors · 2024-08-17
>
> **Q1:** Model trained only on synthetic dataset cannot generate meaningful results due to limitation of dataset diversity, while the length of the synthetic dataset is almost equal to the real dataset. **Can you show some metrics related to the diversity or the distribution for the real-world dataset and synthetic dataset?**
>
> **A1:** Thanks for your suggestion. We show detailed diversity statistics in the following Table. For the reason that model trained only on synthetic dataset cannot generate meaningful results, our explanation is that the RGB data distribution from the rendering engine has a slight gap with real images or videos. Yet, pretraining model used in our pipeline (SD 1.5 and AnimateDiff v3) commonly pretrain their model on large scale real data. Thus, it is hard to finetune them to be controllable diffusion model only with synthetic data. In our practice for training models only with synthetic data, the Stage 1 training (generating a single image from novel human pose and camera pose, based on the reference image sampled from the same video) could generate meaningful images, although its quality is much worse than results trained on real videos. But the Stage 2 training (further finetuning temporal modules to enable video generation) will suffer from large loss functions and cannot converge. Although this paper’s main focus is the accurate camera pose annotations for camera-controllable human image animation from synthetic data, we empirically find that high-quality real videos (even with noisy camera pose parameters) is still vital for training a video diffusion model. In the real data part, our major contribution is that we enable the human image animation model trained on moving camera real videos for the first time by leveraging a camera encoder module. Our method greatly alleviate the need of high-quality static camera videos, as the moving camera videos are easier to collect in a large scale compared with static camera videos.
>
> | Dataset Split | #Subject | #Subject per video | #Motion | #Scene | Avg Clip Length |
> | --- | --- | --- | --- | --- | --- |
> | Internet videos | 24012 | 1.21 | 24012 | 19688 (= #video) | 16.65s |
> | Synthetic (SMPL-X) | 271 (body shape) X 100 (skin textures) X 1,691 (clothing textures) | 1 | 2311 | 200 (HDRI Images) + 587 (locations from a city-scale outdoor scene) | 6.34s |
> | Synthetic (Anime) | 2,381 anime assets | 1 | 40 | 200 (HDRI Images) +  93 (3D scenes by designers) | 3.2s |
>
> **Q2: The rendering quality for synthetic (Anime) data is not good. I suspect that this low-quality data is not helpful for good performance.**
>
> **A2:** Thanks for this detailed suggestion. The major contribution of synthetic data in our model training is the accurate camera pose annotations. Thus, we only utilize the synthetic data to finetune our camera pose encoder and temporal modules, not the 2D appearance related modules (i.e., denoising unet and reference unet). By using such training strategy, leveraging more synthetic data will make our model’s ability non-decreasing with the increase of data size. There is no worry about some synthetic data could have lower quality than the real Internet data, because the appearance part is only trained by the Internet data part. Our training strategy could also be quantitatively supported by our ablation study, where the analysis of quantitative results could be found in Q4.
>
> Besides, in addition to the HDRI images as background, we also have 3D scenes designed by professionals for synthetic videos of Anime characters. The appearance of such cartoon 3D scenes will be better aligned with the Anime characters’ appearance. We hope that such appearance could extend the domain of our HumanVid dataset. We have added such video samples in the ‘Anime_characters/3d/generated_video_1’ subfolder in the cloud drive provided in Line 77 of our Appendix or [this link](https://mycuhk-my.sharepoint.com/:f:/g/personal/1155189552_link_cuhk_edu_hk/EjLaZjITC79Cvqq3uyimlnoB9vSvAhz35b8Fhykt-QHiNw) with password 'humanvid_cuhk'.
>
> **Q3: It seems to be not very clear about the dataset split used to train the model for comparison.**
>
> **A3:** Thanks for the suggestion. We have conduct detailed ablation study to reveal the effectiveness of dataset splits in training. Please refer to the quantitative results in Q4.

---

> > ### Author Rebuttal · Authors · 2024-08-17
> >
> > **Q4: I'm not very convinced that the animation performance with static camera will improve by training with additional low-quality synthetic data.**
> >
> > **A4:** The good performance on static camera is improved by both Internet videos and synthetic with moving camera. In practice, mainly train the appearance branch (Denoising UNet and reference UNet) with the real part of our dataset. When we utilize synthetic data in our model training, only pose guider, camera encoder and temporal modules are trained. Therefore, even some synthetic videos are possible to be not as good as 4K videos filmed by professionals, such data will not have a negative impact on our video diffusion model. The major contribution of training with synthetic data is more accurate human pose and camera pose encoders.  In the following Table, we perform a detailed ablation study to investigate the training strategy of utilizing synthetic data. It shows that using synthetic data only for finetuning camera encoder achieves the best performance.
> >
> > | TikTok test set | Real data in Stage1 & Stage2 | Stage 1 w/ Synthetic data | Stage 2 w/ Synthetic data | SSIM↑ | PSNR↑ | LPIPS↓ | FVD↓ | FID↓ |
> > | --- | --- | --- | --- | --- | --- | --- | --- | --- |
> > | Variant 1 of CamAnimate | ✓ | X | X | 0.677 | 15.957 | 0.333 | 1066.9 | 53.08 |
> > | Variant 2 of CamAnimate | ✓ | ✓ | ✓ | 0.734 | 17.339 | 0.287 | 980.3 | 56.32 |
> > | Ours - final model | ✓ | X | ✓ | 0.778 | 18.762 | 0.247 | 691.8 | 48.36 |
> >
> > High-quality human videos with static camera are very difficult to collect in a large scale, yet it is easier to collect videos without restrictions on camera. With the additional camera parameter encoder, our baseline model is able to learn human animations from videos with moving camera, and control the camera movement precisely in the generated videos. On the contrary, previous methods lack the ability of camera control and thus unable to exploit this type of videos. Once trained, the baseline model is able to achieve good human animation performance in both static camera setting and moving camera setting. Therefore, the synthetic data is not the only source for the good performance of our model. In Table 1 of our Appendix, we quantitatively show that the human appearance is already learnt well by only using Internet videos. The major contribution of our synthetic videos is the accurate camera parameters, which leads to better temporal smoothness and the alignment between videos and camera conditions.
> >
> > Besides, although our major contribution is a dataset and a baseline for camera-controllable human image animation task, an additional benefit of our paper is that we can more easily collect data with moving cameras to train models for static camera scenarios. To the best of our knowledge, no previous methods have tried to utilize videos with moving cameras to train models for static camera human animation.
> >
> > **Q5: If the light and shading effects are considered for the synthetic (SMPL-X) dataset.**
> >
> > **A5:** Thanks for your suggestion. Yes we have considered the light and shading effects in our SMPL-X synthetic data. In Unreal Engine, we use the Sky Light with specifying the environment map, which is an HDRI panorama image, and enabling the shadow casting. The background scene and foreground character share the same light source. In Figure 1 of the attached PDF, we provide light and shading effects of existing data in both the HDRI and the 3D scene, showcasing explicit shadows. In the future, to enhance the model's shadow modeling capabilities, we could add a Directional Light (a function in Unreal Engine) to the scene. As demonstrated in Figure 2 and videos, the Directional Light can be controlled flexibly, which isn't feasible with real data. We will scale up this setup to improve our dataset in the future. The videos showing different directional lights could be found in 'For_Rebuttal/Light_condition' folder in our [cloud drive](https://mycuhk-my.sharepoint.com/:f:/g/personal/1155189552_link_cuhk_edu_hk/EjLaZjITC79Cvqq3uyimlnoB9vSvAhz35b8Fhykt-QHiNw) (password: humanvid_cuhk).
> >
> > **Q6: The trajectories of the camera seem not to be diverse enough. Is it possible to generate the camera trajectory more naturally?**
> >
> > **A6:** We will explain the effectiveness of our camera trajectory from two aspects.
> >
> > (1) As our goal is to train a controllable video diffusion model conditioned on camera trajectory and human pose, we think that let the model be trained on a large range of camera movements (i.e., offsets) in all 6 dimensions (XYZ, roll, yaw, pitch) is important. Yet, the diversity within each individual camera trajectory is not as important as the total covered region in the 6DoF camera pose space by the entire dataset. In Figure 3 of the attached PDF, we show statistics of camera pose offsets of these 6 dimensions. Quantitative results shows that the frame-wise camera pose offset (i.e., the camera movement of next frame) of synthetic data part shows larger distribution than the real data part. It illustrates that our camera trajectory design produces large enough camera movement for modeling natural camera movements in the Internet videos. Besides, experimental results on in-the-wild cases in inference shows that model trained on such camera trajectories could follow camera trajectories in real world videos very well.
> >
> > (2) The current trajectory of camera is achieved by setting two keyframes: the first frame and the last frame. Although two keyframes could already enable the camera movement to cover a large region as we mentioned before, we show some cases using three keyframes and five keyframes for more complex camera trajectories in the `For_Rebuttal/More_complex_camera_trajectory' subfolder in our [cloud drive](https://mycuhk-my.sharepoint.com/:f:/g/personal/1155189552_link_cuhk_edu_hk/EjLaZjITC79Cvqq3uyimlnoB9vSvAhz35b8Fhykt-QHiNw) (password: humanvid_cuhk). We hope that such complex trajectories could be regarded as 'natural' by the reviewer.

---

> > > ### Author Rebuttal · Authors · 2024-08-17
> > >
> > > **Q7: Limitation part:  The error of the camera pose may affect more on the performance. Can you try to quantitatively show the overall error of the camera pose?**
> > >
> > > **A7:** Yes, we agree that the error of the camera pose may affect more on the performance. It is also the major motivation for us to collect synthetic human videos. For the Internet videos, we adopt an off-the-shelf SLAM method to estimate the camera pose since we do not have the ground truth of camera poses in the Internet videos. Since we did not finetune such model, the quantitative results could be found in DPVO’s original paper [57], e.g., Table 1-3 reports results on TartanAir, EuRoC MAV and TUM-RGBD dataset. The average error is 0.21m, 0.105m and 0.089m, respectively. We empirically find that camera parameters estimate from DPVO is useful for training video diffusion models, although it could have errors.
> > >
> > > **Q8: The quality of the synthetic dataset is still limited. It should also be discussed in the paper.**
> > >
> > > **A8:** Thanks for your suggestion. The number of asset and background scenes in the synthetic part is limited when we compare it to real videos. The rendering quality is also certainly worse than the real videos captured by professional cameras. The major advantage of synthetic data is the accurate camera and human pose annotations, and more diverse camera trajectories. We will add descriptions about synthetic data’s limitation to our paper in the final version.

---

> > > > ### Comment · Reviewer_Egiq · 2024-08-18
> > > > **Response to authors**
> > > >
> > > > Thank you for your response. All my concerns have been resolved. I will increase my rating by one.

---

> > > > > ### Author Response · Authors · 2024-08-18
> > > > > **Thanks for your response**
> > > > >
> > > > > Thank you for your thoughtful review and valuable suggestions, which have greatly improved the quality of this work.

---

### Official Review · Reviewer_8Lug · 2024-07-24
**Review of HumanVid**

**Rating:** 7
**Confidence:** 4
**Correctness:** Yes.
**Clarity:** Yes, the paper is well written and ea…

**Review:**

The quality, clarity, originality, and significance of this work is great from my side.

**Strengths:**

1. The task of camera-controllable human image animation is essential, particularly for movie production. It is commendable that the authors excel in this task, proposing both a baseline model and a high-quality dataset.
2. The proposed data collection and annotation pipeline is scalable and valuable, likely to inspire other researchers and facilitate advancements in human image animation.
3. The authors' consideration of licensing and copyright during real video collection ensures that the proposed datasets can be used publicly for both commercial and research purposes.

**Additional Feedback:**

1. Figure 5 shows some visual results of the competitors, including Champ. As I understand, the open-sourced implementation of Champ only supports single-person animation. How were the results obtained in your study?
2. Please report the results on the UBCFashion dataset.

**Documentation:**

Yes.

**Ethics:**

No.

**Limitations:**

Yes, the authors have discussed the limitations of their work.

**Opportunities For Improvement:**

1. The data collection and annotation pipeline needs clearer and more detailed explanations. For example, in the Internet video filtering process, how was it determined whether individuals were occluded or not? Additionally, more details are needed about the human pose fitting process.
2. The lack of breakdown experiments is notable. For instance, while the proposed method outperforms state-of-the-art methods on human videos with camera movements (as shown in Table 3), it is unclear whether the improvement is due to the model design or the curated dataset. A detailed ablation study would help decouple the contributions of the model design and the dataset, thereby strengthening the paper.

**Relation To Prior Work:**

Yes, the differences from previous works are clearly discussed.

**Summary And Contributions:**

This work introduces a large-scale human video dataset for human image animation. It presents a rule-based filtering strategy to collect high-quality real videos from the internet and a comprehensive creation pipeline for generating synthetic human videos. Additionally, it proposes a baseline model for camera-controllable human image animation. Experiments on various datasets demonstrate the effectiveness of the proposed datasets.

---

> ### Author Rebuttal · Authors · 2024-08-17
>
> **Q1:** Data collection and annotation pipeline needs clearer and more detailed explanations. **(1) In the Internet video filtering process, how was it determined whether individuals were occluded or not? (2) what is the human pose fitting process?**
>
> **A1:**
>
> **(1) Occluded individuals.** As we need to process large scale video data, it is hard to accurately remove videos with occlusion. In the practice, we empirically find an effective way to identify occlusions by using the human pose detector. The motivation is that human pose detector tend to fail with occlusion. Thus, if the pose detector produce inconsistent human poses across different frames within a video, occlusions (or some other hard cases, such as uncommon poses or human moving out of the camera view) is very likely to happen. We empirically regard these cases as occluded individuals or hard pose cases that are not suitable for our training set. In the implementation, we compute the average number of frame-wise human poses among all frames as $n$ and keep the video with abs($n$ - round($n$)) < 0.01 as our training set.
>
> **(2) Human pose fitting process.** As the accuracy of 2D human keypoint detection is commonly better than 3D human pose, we adopt the original human pose extraction process in [Animate Anyone](https://github.com/MooreThreads/Moore-AnimateAnyone) to extract 2D human keypoints in the camera space with DWPose [68] and visualize them as the OpenPose format.
>
> **Q2:** Breakdown experiments: whether the improvement is due to the model design or the curated dataset? **A detailed ablation study would help decouple the contributions of the model design and the dataset**.
>
> **A2:** Thanks for your constructive suggestions. We conduct an ablation study on the camera encoder part of our baseline model, i.e., our baseline compared to the [Animate Anyone’s third-party implementation](https://github.com/MooreThreads/Moore-AnimateAnyone), both trained on our dataset. Since most videos in our dataset have moving cameras, it is confusing for video diffusion models to model both camera movement and human movement only conditioned on human poses. Thus, the performance of two models trained on the same training set shows significant difference. In the following Table, we report these two model’s performance on the TikTok dataset test set. Our model outperforms animate anyone by a large margin. It shows that the camera condition and its encoder in our proposed baseline model to provide necessary information to the video diffusion model is important and reasonable. Since previous methods trained on moving camera videos commonly cannot achieve good performance, our model design is also a contribution to this area, i.e., it enables the good performance on static camera videos by training models on moving camera data for the first time.
>
> | TikTok test set | SSIM↑ | PSNR↑ | LPIPS↓ | FVD↓ | FID↓ |
> | --- | --- | --- | --- | --- | --- |
> | Animate Anyone | 0.658 | 15.954 | 0.337 | 1133.1 | 53.65 |
> | Ours | **0.778** | **18.762** | **0.247** | **691.8** | **48.36** |
>
> In addition to the contribution of our model design, our dataset provides high-quality videos with diverse human appearance (mainly from the real data part) and accurate camera and human pose annotations (mainly from the synthetic data part). It further contribute to this area by supporting the model training of movie-level human-centric video generation with moving cameras. For the ablation study of the real part and synthetic part of our dataset, please refer to Table 1 of our Appendix.
>
> **Q3: Champ only supports single-person animation.**
>
> **A3:** As previous methods and our method commonly adopt the human pose visualization as the pose condition, all methods are able to process non-overlapping multi-person scenes in principle. Because the image-like pose condition could have multiple pose visualizations in a single feature map. In the original scripts of [Champ](https://github.com/fudan-generative-vision/champ), the 3D pose detector 4D-Humans [a] actually produces multiple human poses. Yet, their blender scripts in rendering depth and normal only accepts one human pose. We enable Champ to handle multiple humans by a simple adjustment on their blender rendering scripts. This script will also be publicly available when we release our code. The visualizations of rendered depth, normal and mask are shown in the PDF. The multi-human video generated by Champ could be found in the 'For_Rebuttal/Multi-human_Champ' folder in the [cloud drive](https://mycuhk-my.sharepoint.com/:f:/g/personal/1155189552_link_cuhk_edu_hk/EjLaZjITC79Cvqq3uyimlnoB9vSvAhz35b8Fhykt-QHiNw) (password: humanvid_cuhk).
>
> **Q4: Results on UBCFashion dataset.**
>
> **A4:** In the following Table, we show the comparison with previous methods on UBC Fashion dataset. The evaluation protocol is the same with descriptions in Sec 4.1 of our paper, i.e., testing all 24 frames generated by methods with stride 3, covering the content of frame [1,72).
>
> | UBC Fashion Test Set | SSIM↑ | PSNR↑ | LPIPS↓  | FVD↓  | FID↓  |
> | --- | --- | --- | --- | --- | --- |
> | Magic animate* | 0.6027 | 6.6634 | 0.5526 | 1583.9 | 118.7692 |
> | Animate Anyone | 0.9140 | 23.1637 | 0.0698 | 345.4 | 33.7706 |
> | Champ | 0.9228 | 25.2692 | 0.0575 | 269.4 | **27.3567** |
> | Ours | **0.9296** | **25.9215** | **0.0498** | **256.6** | 29.3018 |
>
> *Since the training code of Magic animate is not released, we directly use the [official inference code and checkpoint](https://github.com/magic-research/magic-animate) to inference on UBC Fashion dataset. According to the information provided in the Github, this checkpoint is trained on TikTok dataset. Thus, it cannot ensure a clean background in the UBC Fashion test set, leading to much worse performance than other methods.
>
> [a] Goel, Shubham, et al. "Humans in 4D: Reconstructing and tracking humans with transformers." *Proceedings of the IEEE/CVF International Conference on Computer Vision*. 2023.

---

### Official Review · Reviewer_KjST · 2024-07-28
**Interesting work for the human animation community**

**Rating:** 6
**Confidence:** 2
**Correctness:** Yes.

**Review:**

### pros
- The datasets presented here are made publicly available and could facilitate research in the field.

### cons
- Grammar, flow, structure, and readability should be improved

**Strengths:**

- The authors consider for their synthetic dataset different body shapes and skin tones, clothing and textures, as well as anime characters.
- The make their datasets publicly available which would facilitate research in human animation
- The authors use both a quantitative and qualitative assessment, including a user study

**Additional Feedback:**

NA

**Clarity:**

Clarity needs to be improved. The language, grammar, structure, and flow of the manuscript need improving

**Documentation:**

Yes.

**Ethics:**

No.

**Limitations:**

The authors address limitations and potential negative societal impacts.

**Opportunities For Improvement:**

- Do the authors make the annotations publicly available?
- It does not seem like the code was made publicly available
- There is not much info on how SLAM is applied

**Relation To Prior Work:**

Yes.

**Summary And Contributions:**

The authors introduce a synthetic and real-world dataset for human animation.

They also produce a pipeline for human video generation that is able to produce annotations for human and camera motions.

---

> ### Author Rebuttal · Authors · 2024-08-17
>
> **Q1: Paper writing should be improved.**
>
> **A1:** Thanks for your suggestion. We will carefully revise our paper in the final version.
>
> **Q2: Details about data release.**
>
> **A2:** Yes. We will make the data, annotations and code publicly available along with our paper’s final version.
>
> **Q3: More information on slam.**
>
> **A3:** We use the official implementation of DPVO [57] in the [Github repo](https://github.com/princeton-vl/DPVO) to extract camera extrinsic. In DPVO, the camera intrinsics need to be set as hyper-parameters. We empirically set such values for Internet videos (e.g., focal = 35mm for landscape videos and focal = 75mm for portrait), and our experiments show that camera extrinsic parameters estimated from such intrinsic parameters is effective for training diffusion models.

---

### Author Rebuttal · Authors · 2024-08-17

Dear Reviewers,

We sincerely appreciate your time and effort in reviewing our paper. Your constructive feedback has been invaluable to us. In response to your comments and to provide a more comprehensive view of our work, we have created a dedicated website showcasing additional qualitative results from our model. We invite you to explore these results at [this url](https://humanviddataset.github.io/).

We have carefully addressed the specific concerns raised by each reviewer, and our detailed responses can be found below each review. We hope that our explanations and the additional qualitative results in the website could effectively address your queries and provide clarity on any points of concern.

Thank you once again for your thorough evaluation and insightful suggestions. Your input has been crucial in refining and improving our research.

Sincerely,

The Authors

---

> ### Author Response · Authors · 2024-08-26
>
> Dear Reviewers,
>
> We are happy to address any additional concerns you may have. If, after our discussion, you find that all your concerns have been resolved, we would greatly appreciate your consideration in updating the final rating.
>
> Thank you for your time and expertise.
>
> Sincerely,
>
> The Authors

---

### Comment · Area_Chair_1c3Z · 2024-08-28

Dear Reviewers,

Thank you for taking the time to review this submission. :)

This is a gentle reminder regarding the reviewer-author discussion.

Please respond to the author's rebuttal at your earliest convenience, especially if you have any points of disagreement.

The deadline is August 31 at 11:59 PM AoE!

Early discussion is always appreciated.

Best, AC

---

### Decision · Program_Chairs · 2024-09-26

**Decision:**

Accept (Poster)

**Comment:**

We appreciate the authors' efforts in answering reviewers' questions during the rebuttal phase and to a great extent addressed the reviewers' concerns. The paper received diverse ratings: 6,7,7,7. The reviewers are generally positive about the proposed idea and the problem considered to synthesize 2D human animation generation with camera motion. The decision takes into account the paper, the reviews, the rebuttal, and the post-rebuttal reviewer discussion. The AC panel did not find any reason to overturn the majority rating. Based on the above consideration, the AC suggests to accept this paper but the revised version needs to properly include the responses to Reviewers' comments, where possible, the workable links to the dataset and the code.